# Telocytes and Macrophages in the Gut: From Morphology to Function, Do the Two Cell Types Interact with Each Other? Which Helps Which?

**DOI:** 10.3390/ijms23158435

**Published:** 2022-07-29

**Authors:** Maria Giuliana Vannucchi

**Affiliations:** Department of Experimental and Clinical Medicine, University of Florence, 50139 Florence, Italy; mariagiuliana.vannucchi@unifi.it

**Keywords:** PDGFRα-positive cells, interstitial cells, immune cells, cell contacts, transmission electron microscopy, immunohistochemistry, cell recruitment, nursing cells

## Abstract

Telocytes and macrophages are ubiquitous cells located in loose connective tissues and share the same mesenchymal origin. Despite these common elements, depending on where they reside, these two cell types are profoundly different in terms of their morphology and functions. The purpose of this review is to provide an update on the knowledge regarding telocytes and macrophages in the gut, where their presence and significance have long been underestimated or misunderstood. The focus will be on the possibility that these two cell types interact with each other and on the potential meaning of these interactions. Based on the complexity of the topic, the variety of possible methodological approaches and the expertise of the author, the point of view in the discussion of the literature data will be mainly morphological. Furthermore, considering the relatively recent period in which these cell types have acquired a primary role in gastrointestinal functions, the attention will be greatly confined to those articles published in the last decade. The microbiota, another main protagonist in this context, will be mentioned only in passing. It is hoped that this review, although not exhaustive, will highlight the importance of macrophages and telocytes in the complex mechanisms that ensure intestinal functions.

## 1. Introduction

The gut wall is an elaborate structure in which all of the mammalian tissues are represented. These are organized into three main layers: The muscularis externa, covered by the serosa, is made up of smooth muscle cells arranged in two layers, parallel (longitudinal) and perpendicular (circular) to the major axes of the organ. In between these layers, a complex neural plexus, named the myenteric or Auerbach’s plexus, is present, which consists of ganglia and nerve strands. Going inward, there is the submucosa, a loose connective tissue containing the other neural plexus named the submucosal or Meissner’s plexus, which is formed by smaller ganglia and thin nerve bundles. The submucosa is delimited, in depth, by the muscularis mucosae, a subtle layer of smooth muscle cells beyond which there is the mucosa. The latter is a complex layer formed by a loose connective tissue very rich in cells and nerve fibers, called the lamina propria (LP), and by the epithelium made up of a covering layer and glands.

The two neural plexuses are connected to each other though nerve bundles that cross the gut wall in a transversal direction. Nerve fibers from the myenteric plexus (MP) innervate the muscularis externa and regulate gut wall motility; those from the submucous plexus (SMP) reach the apex of the villi taking contact with vessels, interstitial cells and epithelial cells and regulate absorption and secretion [1].

All of the cells forming these layers are, directly or indirectly, related to each other to ensure the functionality of the organ. Among these, some cells, such as the neurons, are often considered the main characters while others have a minor role. This axiom is progressively changing, and, in the last decade, other cells have attracted the attention of different research groups [2,3,4]. In this review, the focus will be on two types of connective cells, telocytes (TCs) and intestinal (resident) macrophages (MCs), highlighting their morphological properties and assessing the possibility of interaction with each other and whether this interaction affects their ability to guarantee intestinal functions.

The name *telocyte* entered scientific literature in 2010 [5] to indicate a ‘*new*’ cell type present in the interstitium of almost all organs examined [6,7]. TCs are cells of mesenchymal origin, show a peculiar shape and play several roles depending on their location.

*Macrophages* share the same mesenchymal origin of the TCs and are commonly found in loose connective tissues. Two exceptions are known: the microglia located in the central nervous system and the dendritic cells, or Langerhans cells, located in the epidermis [8]. Macrophages are immune cells that exhibit different features and play distinct roles depending on local and systemic stimuli [2,3,8,9].

## 2. The Gut Telocytes

### 2.1. Morphological Features and Immunohistochemical Properties

TCs’ identification is primarily due to a Romanian research group headed by Professor Popescu [5]. In the gut, TCs were initially superimposed to the interstitial cells of Cajal (ICCs); however, doubts about a common identity forced researchers to distinguish the TCs from the ICCs, and, in the absence of a not-yet convincing identification, they were firstly named interstitial Cajal-like cells (ICLCs). Other authors underlined the similarities between these cells and fibroblasts and called them fibroblast-like cells (FLCs) [4]. To stop potential confusion during these cells’ identifications, Popescu et al. [5], using a transmission electron microscope (TEM), established the ‘*ultrastructural hallmarks*’ that characterize all TCs [4,5], and the impressive number of papers later published on the topic of TCs confirmed that only the TEM certifies the identity of TCs with certainty [4,5].

Conversely, when studied using light or fluorescent microscopy, a consistent variability of TCs’ immune phenotype was reported. This variability was attributed to the ability of TCs to adapt to the different sites in which they reside, to the different functions they possibly accomplish and, finally, to the presence of TCs’ subtypes [4]. In the gut, unlike in other organs, TCs express both CD34 and PDGFRα and never express c-Kit. These peculiarities make these TCs easily distinguishable from the ICCs and identifiable with a relative certainty using immunohistochemistry, particularly with PDGFRα immunolabeling [10,11,12].

### 2.2. Locations in the Gut Wall

Since the paper by Popescu and Faussone-Pellegrini [5] was published, these cells have been described in all layers of the gut where they come into contact with each other through their long processes to make homotypic networks. In the muscle coat, the TC network runs parallel with that made by the ICCs and forms a three-dimensional (3D) net that covers the border between the circular muscle and the submucosa [10,11,13,14,15]. Then, unlike the ICCs, the TCs extend their network into the submucosa, encircle the ganglia, make tight meshes in the loose connective tissue, embrace the muscularis mucosae and terminate within the mucosa, forming a 2D network bordering the glands and the epithelium [4,11,13]. All over, the TCs run close to almost all of the cells present in the gut interstitium, such as ICCs, fibroblasts, immune cells, neural cells, smooth muscle cells and blood capillaries, and make direct contacts with some of them.

### 2.3. Cell-Interactions and Roles

The ability of the TCs to form networks and make contact with other cell types likely conditions the numerous proved or theorized roles attributed to these cells in the gut. Through the 3D network, during peristalsis and regional movements, TCs ensure an appropriate distribution of the mechanical stresses in the submucosa and mucosa and guarantee the spatial organization of the two layers, maintenance of the correct localization and support for the ganglia and nerve strands; extra cellular matrix (ECM); cells, such as fibroblasts or immune cells; and vessels [4,16]. Moreover, some studies [17,18] provide evidence of their capabilities to correctly orient the ECM components once synthesized by the fibroblasts, and, under hormonal stimuli, the TCs themselves synthesize the ECM [19].

Convincing data have been accumulated to demonstrate that the TCs that line the epithelium are nurses for the cryptal stem cells [4,20,21]. In agreement with this role is their PDGFRα receptor positivity as PDGF/PDGFR signaling has proved crucial for intestinal villous morphogenesis [22]. Genetic studies [23,24] designated the TCs lining the crypts as regulators of local stem cell proliferation and those lining the apex of the villi as mediators of cell differentiation. Interestingly, by combining the genetic [23,24] and morphological data [11,13] it has been assumed that TCs condition the differentiation, rather than proliferation, of epithelial stem cells [4].

In the gut, as in other organs, TCs can interact with other cells, either by establishing cell membrane contacts [4,25] or producing exosomes, i.e., vesicles of different size, which operate as intercellular shuttles that deliver chemical signals [4,19,26]. The cell contacts are mainly located along their long and thin processes and might act as mechanical cell-to-cell attachments or sites of intercellular communication [7,13,14]. Of note, extensive appositions of the plasma membrane and the presence of small exosomes were found between the TCs and immune cells [7,25,26,27,28]. In this regard, Kurahashi et al., [11] reported the presence of the toll-like receptor 4 and 5 genes in mouse subepithelial TCs. Moreover, TCs, through the cell contacts they have with the ICC, may take part in the pace-making activity or neurotransmission intermediation played by the ICC [4,13,14,29,30]. Gut TCs also establish contact with the smooth muscle cells [4,29,31,32], and, being often interspersed with the ICCs, the existence of an integrated circuit encompassing SMCs, ICCs and TCs, the *SIP syncytium,* has been proposed in order to guarantee the correct function of the muscle coat [29,30,31]. Although cell contact between TCs and neurons was never observed under the TEM [25,28,33], TCs express neuronal markers [15,29,30] and respond to some neuronal mediators [32]. Finally, in contrast with some reports on the heart [34] and sensitive ganglia [35], no data are available on the presence of contact between TCs and glial cells in the gut.

## 3. The Intestinal (Resident) Macrophages (MCs)

### 3.1. Morphological Features and Location

In the GI apparatus, a rich and variegated population of resident macrophages (MCs) is present. The high number of MCs is not surprising given that the intestine is the largest reservoir of immune cells in the body. A significant contribution in identifying the features of intestinal MCs was provided by Mikkelsen and coworkers [36,37], who described these cells in the muscle walls of the small and large intestines of humans and mice using both light and electron microscopes. In both humans and rodents, MCs populate the intestine along its entire length, reaching their highest density in the colon [38]. Moreover, MCs, although distributed throughout all layers of the intestinal wall, show important differences in their densities [28], shapes and times of replenishing [28,38,39,40] according to their location [28,41,42]. Notably, in mice, the density of the MCs is high in the proximal and distal colon, while it is consistently lower in the transverse colon (personal observation).

In the mucosa, the loose connective tissue that forms the lamina propria (LP) harbors a rich population of MCs in terms of both the number and cell variety. Those located along the axis of the villus are mostly spheroidal in shape, while those adjacent to the epithelium and the glandular crypts, which contain stem cells and Paneth cells, are rich in thin, short, ramified processes [8,42].

In the submucosa (SM), the MCs present several processes and are closely associated with the numerous blood vessels and the SM ganglia [8,42].

In the muscle wall, the MCs are mainly positioned in close proximity to the MP [8,36,37,42,43], forming an intricate network with other connective cells, such as the ICC and TCs. These MCs are richly ramified, while a few scattered, thin, elongated MCs are present among the muscle fascicles and within the serosal layer [28,36,37]. The MCs’ distributions results were similar in the guts of both the rodents and humans [28,36,37].

### 3.2. The Ontogenesis

All MCs derive from mesenchyme; however, as they belong to the hematopoietic cell line, their behavior during ontogenesis varies according to the site in which they reside and their fate. Thus, the MCs’ ontogenesis deeply influences the wide variety of shapes, cellular interactions and performances of MCs.

The primitive intestine derives from the yolk sac that represents the first site in which the hematopoiesis takes place (during the third week of gestation in humans and around E7.5 in mice) [8,42]. Therefore, it is not surprising to find out that the first macrophage population present in this organ is constituted by hematopoietic embryonic precursors with the capacity for self-renewing [8,42]; notably, these precursors produce primitive erythrocytes and MCs (but not monocytes) [8]. A similar fate is described for those MCs that enter the central nervous system during development to become microglia [8]. The main difference between the two sites is that the intestine does not form a blood barrier to impede other more differentiated macrophage precursors from entering the gut wall. Indeed, with time, circulating monocytes originating from bone marrow also colonize the intestine, as demonstrated after MCs’ lethal irradiation and bone marrow transplantation [2,3,42,44]. In the adult intestine, as in other organs, MCs consist both of cells rapidly replaced by incoming monocytes and long-lived self-maintaining cells [3,38,42,43,44]. This finding disproves previous beliefs that the MC pool depended solely on the replenishment and differentiation of incoming monocytes [3,42]. Of note, in the muscularis externa and submucosa, the portion of long-lived MCs is predominant, whereas, in the LP, the MCs subjected to high levels of turnover prevail [3]. Finally, in the LP, it has been reported that the long-lived MCs originate from both embryonic and bone marrow cells, thus indicating that longevity is not an exclusive property of embryo-derived macrophages [3,38,45].

### 3.3. Cell-Interactions and Functions

In the gut, the MCs reside permanently, share an anti-inflammatory phenotype and exert phagocytosis [42]. Moreover, beyond mediating the innate immunity, intestinal MCs are involved in intestine homeostasis and functionality by interacting with other connective tissue cells [4,28,36] and with the neural cells [2,28,43,44]. The intestinal MCs are divided into subpopulations, each of which shows an activity strictly related to the anatomical niche (the layers) that it occupies, and this resulting functional heterogeneity is essential to guarantee intestinal integrity.

The LP-MCs surveil the environment, phagocytose potentially harmful antigens or apoptotic cells [44], respond to pathogens present in the lumen or crossing the epithelium and sample the presence of antigens in the lumen through dendritic projections that across the epithelial barrier [42,43,46,47]. Once the antigen is trapped, it passes to the CD103+ dendritic cells, which, in contrast with the MCs, have a migratory property and enter lymph nodes to prime the T cells [48]. The LP-MCs that line the glandular crypts are likely involved in the epithelium’s proliferation and differentiation, as their depletion impairs the epithelium’s renewal [8,42,43,48,49]. Remarkably, despite the highly phagocytic nature of the LP-MCs, their interaction with the commensal microbiota or food antigens does not lead to an inflammatory response [42,46]. This is due to the expression of the anti-inflammatory cytokine IL-10 and its IL-10 receptor (R), which make these cells tolerant towards harmless commensal bacteria and food antigens, a crucial condition for the organism’s survival [38,42,43]. The IL10-IL10R axis has been described in both humans and rodents, and its disruption implies an expression of higher levels of pro-inflammatory mediators by intestinal MCs, thereby leading to spontaneous intestinal inflammation [50,51,52,53]. The LP-MCs’ presence and activity depend on two cytokines: the colonic stimulation factor (CSF)1, produced by the cryptal cells of the epithelium, and the CSF2, produced by the 3-innate lymphoid cells [8,54,55,56]. In particular, the CSF2 also stimulates the macrophages to produce IL-10 [8,56]. Finally, several pieces of evidence indicate a strict dependence of the intestinal MCs on the microbiota: a significant increase in LP-MCs is observed with weaning, while antibiotics slow down their turnover, and few LP-MCs are found in germ-free mice [38,53,54,57].

Less information is available on the role of the MCs in the submucosa. De Schepper and a coworker [3] demonstrated that most of these MCs are self-maintaining and long-lived phagocytes, similar to what had been reported for those located in the muscularis externa. Furthermore, also in this region, MCs may have a trophic effect as their decrease causes neuronal loss, damage to blood vessels with bleeding and reduced intestinal secretion [3,42].

The muscular (M)-MCs have a privileged interaction with the myenteric neurons. They are fundamental to guaranteeing the neuronal steady-state through apoptotic debris phagocytosis during development and adulthood [42,58]. Moreover, they contribute to maintenance of the gut’s homeostasis by a bidirectional crosstalk with the neurons. In particular, M-MCs produce macrophage-derived bone morphogenic protein-2 (BMP2), which binds to the BMP receptor (BMPR) expressed on the enteric neurons to induce the secretion of CSF1, which, in turn, supports the M-MCs and stimulates the additional production of BMP2 [1,43,58]. It has been reported that the absence of macrophages in CSF1-deficient mice results in an increased myenteric neuronal number and a less organized enteric nervous system (ENS) architecture [2,8]. Furthermore, broad-spectrum antibiotics alter M-MCs-enteric neuron interaction and peristalsis [2,3,38,43], suggesting that the microbiota can regulate intestinal motility as well by acting on M-MCs [2,3,38,43,58,59]. Changes in the M-MCs’ behavior and microbiota composition were described during aging, and both of these factors were associated with alterations in the ENS and intestinal dysmotility [43,44,58,60]. Based on all these data, it has been suggested that the existence of a “triad” composed of ENS, MCs and microbiota, whose cross talk is mediated by different molecules, is essential for regulating intestinal functions [2,3,58]. The possibility that MCs and enteric neurons may form cell contacts has also been considered, and, in recent years, several immunohistochemical studies seem to confirm this possibility [2,3,40,61]. To our knowledge, only three papers have investigated the possibility of performing ultrastructural experiments [28,36,41]. Indeed, the results were contrasting. Two of them [28,36] found no membrane apposition between the two cell types. Moreover, Ji et al. [28] underlined that, in the MP, when close, the distance between the MCs and the neurons or axons was equal or less than 400 nm, a value that cannot be resolved under a light or fluorescent microscope, giving the impression that the two cells are in contact. On the contrary, Dora et al. [41] described cell contact between myenteric neurons and those MCs located inside the ganglia. In summary, considering that, among the roles played by the intestinal MCs, there is also that of identifying and phagocytizing aged or damaged neurons, the question remains unanswered, and further studies are necessary to confirm whether and when these two cell types establish contact.

### 3.4. The Recruitment, the Niches and the Longevity

When the monocytic precursors enter the murine gut, they acquire CX3CR1 expression and lose CCR2, giving rise to fully matured tissue-resident macrophages characterized by the expression of F4/80, CD64, CD163 and CD206. This process requires five to six days, involves major gene expression changes and produces cells with increased phagocytic capacity and constitutive IL-10 production [42,51,52,53]. The macrophage differentiation and maintenance are controlled by diverse molecules, among which the CSF1 and CSF2 have a main role. For example, mice lacking CSF1 or its receptor CSF1R show significant macrophage depletion [2,8,42,62]. Less known are the local factors that condition the longevity of the different intestinal MC populations that promote them towards long-lived or rapidly replenishing cells. Since the percentage of one or the other MC populations greatly vary amongst the gut niches, independently of the fact that each niche contains MCs of both embryonic and bone marrow origin, it is reasonable to assume that there are niches that support macrophage longevity and self-renewal and others that do not. 

In the LP, it is convincing that recruitment is determined by “physiological inflammation” due to the constant exposure to antigenic dietary material or commensal bacteria [38]. The presence of continuous inflammation may also explain the short lifespan of these MCs and their rapid replenishment [3,42]. In summary, these findings confirm that and demonstrate how the LP niche affects the longevity of the MCs.

In the muscularis externa (and in the submucosa), instead, the majority of the MCs are long-lived and self-renewing; therefore, this niche seems to own properties that guarantee long life and/or self-renewal. Studies of transcriptome [3] have shown that the M-MCs are transcriptionally homogeneous, while the LP-MCs show four different transcriptome subpopulations [3]. These results were interpreted as proof that the MCs features are under the control of the niche environment rather than due to intrinsic properties of the cells [3,42,43,48,50,63]. Thus, the questions are: are there cells/molecules that are able to influence the recruitment and longevity of M-MCs? Which are they? As previously reported, enteric neurons play a key role in ensuring the presence of MCs in adequate proportions. However, studies during development indicate that MCs precede neural stem cells in colonizing the intestinal wall [64], thus ruling out the possibility that neural stem cells influence MCs’ recruitment, while MCs are essential for neural differentiation, proliferation and survival [3,42,64].

## 4. Do TCs and Intestinal MCs Interact with Each Other? Which Helps Which?

Little is known about the possibility that other cells can interact with intestinal MCs. Immunohistochemical studies have shown strict spatial relationships between MCs and interstitial cells, such as mast cells and the ICC [36]. However, possible functional interactions between the MCs and these cells have been reported only in pathological conditions [36,65].

### 4.1. The Gut TCs Interact with the MCs

We recently investigated, in human and mouse gut, the possibility that MCs might interact with [4,28]; personal observations]. Our studies showed that these MCs are constantly located in proximity to TCs along the entire thickness of the wall (personal observations). Moreover, under the TEM it was proved that, in both the human and mouse gut, TCs and M-MCs have extended cell-to-cell contact, and, in some cases, the TCs seem to almost encircle the MCs with their long and thin processes [28] (Figure 1 and Figure 2).

### 4.2. Which Helps Which?

TCs-MCs spatial relationship and cell-to-cell contact may have different functional meanings. The two cells’ vicinity is likely related to the scaffold role that TCs play in all organs [4]; however, the existence of cell contacts per se is indicative of direct interactions and the extent of the contact is highly peculiar. As already reported, MCs occupy specific sites, the niches, where they remain for all of their life in order to perform their functions. Thus, through their contact with the TCs, the MCs more easily maintain their position within the TC meshes and better sustain stretching during contractile activity, thereby avoiding dislocations outside the niches. Moreover, through the extended contact, whichalmost encircles the MCs, the TCs might create a microenvironment able to shelter the macrophages from possible noxie or adverse conditions and favor their longevity. Interestingly, the extended contacts were mainly observed in those niches in which the long-lived MCs are the majority, such as the submucosa and the muscularis externa (Figure 1 and Figure 2). In summary, it is conceivable that, in the gut, the TCs affect the location and activities of the MCs and, in some cases, may even act as nurses protecting them from local aggressors.

According to our results, no extended contact was observed between the TCs and the LP-MCs (personal observations). This is not surprising since, in the LP, the TCs change their spatial organization from a 3D network to a 2D network lining the epithelium; a morphological change that is associated with a TC’s function as a nursing cell for the cryptal stem cells [4,20,21,22,23,24]. On the other hand, due to the presence of food antigens, molecules of a different nature and microbes, the LP is the site where MCs primarily play their role as immune cells and where they need to move freely to reach the epithelium or to come into contact with other immune cells. This condition also explains why most of the LP-MCIs have a shorter lifespan and a faster replacement. Interestingly, an exception to this rule is likely represented by the LP-MCs’ proximity to the crypts that would be involved in epithelial proliferation [8,42,43,48]. Consequently, the possibility that these macrophages and the TCs could functionally interact cannot be ruled out.

Finally, in our paper [28], we describe the presence of simultaneous M-MC + TC + SMC cell-to-cell contact, between which the TC is located (Figure 1B). Although this report may coincidentally correspond to those cases in which, by chance, the TC simultaneously plays its mechanical role towards both the SMC and M-MCs, it is possible that, through the TCs, a second anatomical and functional circuit, similar to the one called the *SIP syncytium* by Sander and coworkers [29,30,31], is present to regulate muscle contractility. A schema concerning this putative *TC-MC-SMC circuit* is presented in Figure 3.

## 5. Conclusions

Most of the data presently discussed originate from morphological studies. The choice is not casual, as morphology is a convincing approach to understanding the roles of the numerous cell types present in the gut. Through morphology, in fact, it is possible to distinguish single-cell populations based on their unique features, to identify the niches in which they are harbored and to establish the existence of cellular contact with each other and/or with other cells. This information is essential for assigning, confirming or integrating the functional roles deduced or hypothesized with other procedures.

Concretely, with TEM, it was possible to identify the TCs with certainty and to establish that they are a unique population of interstitial cells. Furthermore, the study of their phenotypes led to distinguishing intestinal TCs from those residing in other systems. Similarly, morphology allowed for description of the intestinal MCs’ distribution in the different niches of the gut wall and for correlation of these findings with the different roles that these cells might play.

Intriguingly, from the data available, it appears that TCs and MCs establish cell-to-cell contact and, likely, interplay in order to perform their functions in the muscularis externa and submucosa, in which both cell types are also directly or indirectly related to neural cells. In this relationship, the TCs often perform a role of support and nursing to guarantee performance of the MCs. On the contrary, in the mucosa, their fate seems to diverge, and each cell type plays a separate function. In fact, in the LP, the TCs form a 2D network lining the epithelium and play a nursing role for the cryptal stem cells, whereas the MCs mainly accomplish their most characteristic role, that of functioning as immune cells.

## Figures and Tables

**Figure 1 ijms-23-08435-f001:**
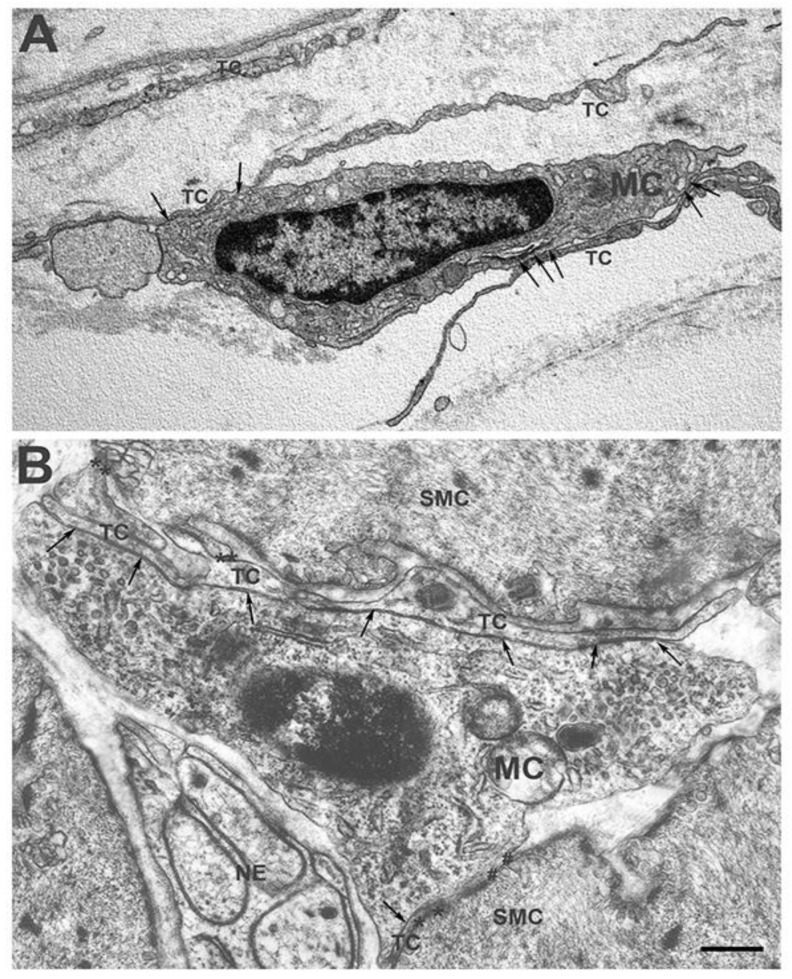
**Transmission electron microscope. Human stomach.** (**A**) Submucosa. Some TCs border an MC with their long processes. Both TCs make extended contact with the MC with their thin processes (arrows). (**B**) Muscularis externa—Some TCs processes make extended contact with an MC (arrows) and with the SMCs (asterisks). The MC also makes direct contact with a SMC (hashtags). These images were reproduced from those presented in the paper [28]. Scale bar = 0.6 μm.

**Figure 2 ijms-23-08435-f002:**
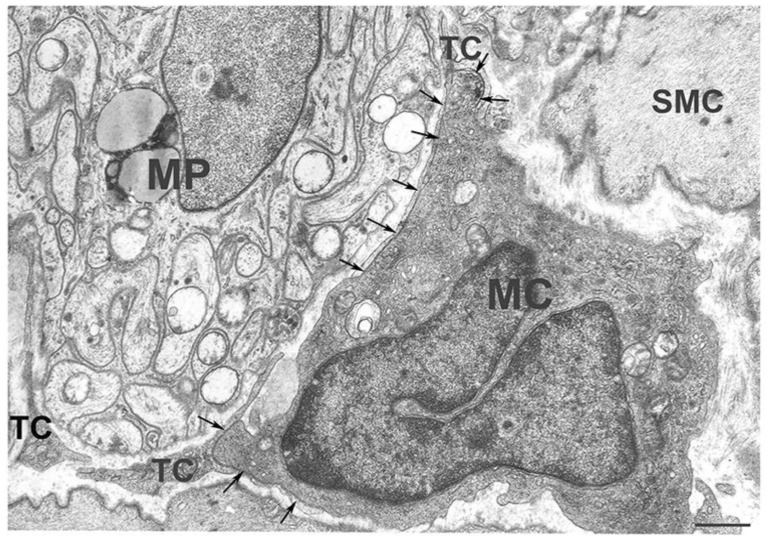
**Transmission electron microscope. Human colon.** Myenteric plexus. Some long and thin TC processes encircle a ganglion and make extended contact with an MC (arrows) The image was reproduced from those reported in the paper [28]. Scale bar= 0.6 μm.

**Figure 3 ijms-23-08435-f003:**
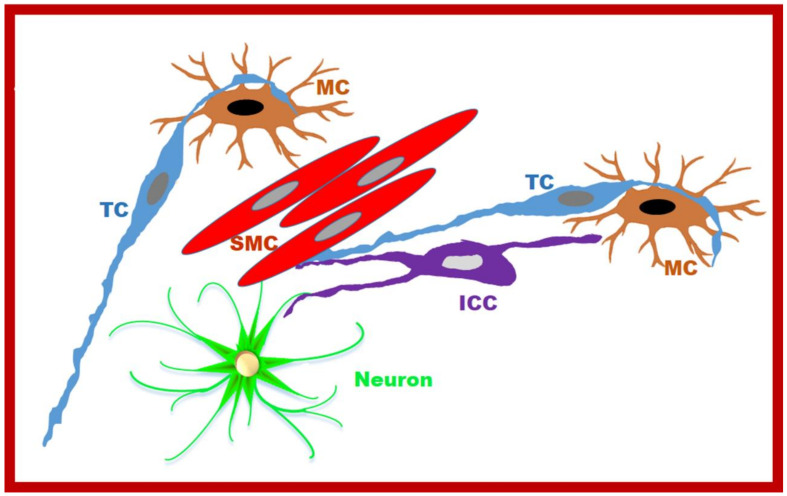
A schema of the complex interaction between the TCs and the MCs in the muscularis externa. On the **left**, an extended point of contact between a TC and an MCs is reproduced. On the **right**, the hypnotized *TC-MCs-SMC circuit* is proposed. Finally, the proposed *SIP syncytium* [29,30,31] constituted by ICC-P-SMC, where P corresponds to the PDGFRα+/TC cells, is also mimicked. The neuron makes contacts with the ICC and SMC.

## Data Availability

The datasets generated for this study can be obtained upon reasonable request by email to the corresponding author.

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
