# Peer review of "Telocytes and Macrophages in the Gut: From Morphology to Function, Do the Two Cell Types Interact with Each Other? Which Helps Which?"

_ijms, 2022, doi:10.3390/ijms23158435_

Round 1

Reviewer 1 Report

 The author described about the gut telocytes (TCs) and intestinal macrophages (IMCs). She precisely showed the detailed description of their features, location, function, and cell-to-cell contacts. Although this review is entirely well-written, the fourth section, “4. Do TCs and IMCs interact with each other? Who helps who?”, mostly consisted of the authors’ experiments and insistence. The structural connections were described precisely. If possible, the explanation or hypotheses for the functional meanings of the cell-to-cell connections would be added.

 In addition, “MP” (at line 43) and “SMP” (at line 44) in the section of 1. Introduction should be spelled out.

Author Response

I though this review is entirely well-written,

I wish to thank the referee for her (his) positive evaluation of the review

.. the fourth section, “4. Do TCs and IMCs interact with each other? Who helps who?”, mostly consisted of the authors’ experiments and insistence. The structural connections were described precisely. If possible, the explanation or hypotheses for the functional meanings of the cell-to-cell connections would be added.

In chapter 4.2 a list of possible functional meanings of the cellular contacts between TC and MC has been reported: i) TCs constitute the scaffolding through which the macrophages are dislocated in the correct site; ii) through cellular contacts, TCs ensure that macrophages stay in the appropriate niche; iii) by surrounding the macrophages, the TCs create a sort of protective shield from any adverse conditions. The term ‘functional’ was added to ameliorate the comprehension of this paragraph

In addition, “MP” (at line 43) and “SMP” (at line 44) in the section of 1. Introduction should be spelled out.

The two acronyms have been spelled out as suggested.

Reviewer 2 Report

In this manuscript, Dr. Vannucchi reviewed the two important mesenchymal derived cell types - telocytes and macrophages, in the gut. The author also made claims about the interaction between  telocytes and macrophages, based on her publications and personal observations. This manuscript provided some functional insights of TC-MC functions in the gut.

Line 43, please define the abbreviation MP in line 36.

Line 44, please define SMP.

Line 85, “since the paper was published” – since the initial discovery paper of TC, or something similar.

Line 132. “intestinal (resident) macrophages (IMC)”. If the abbreviation of IMC is not commonly used by the field, I suggested that the authors just used “intestinal macrophages” instead.

What are the roles of TC and MC in mediating intestine-microbiota interaction? 

Author Response

In this manuscript, Dr. Vannucchi reviewed the two important mesenchymal derived cell types - telocytes and macrophages, in the gut. The author also made claims about the interaction between telocytes and macrophages, based on her publications and personal observations. This manuscript provided some functional insights of TC-MC functions in the gut.

I thank the referee for considering the review as a source of insights

Line 43, please define the abbreviation MP in line 36.

Done

Line 44, please define SMP.

Done

Line 85, “since the paper was published” – since the initial discovery paper of TC, or something similar.

The sentence has been modified to make it more precise

Line 132. “intestinal (resident) macrophages (IMC)”. If the abbreviation of IMC is not commonly used by the field, I suggested that the authors just used “intestinal macrophages” instead.

All IMC acronyms have been replaced with MC. When deemed necessary, the acronym MCs was preceded by the adjective 'intestinal' 

What are the roles of TC and MC in mediating intestine-microbiota interaction? 

As stated in the abstract, the relationship between TC and / or MC and the microbiota is only hinted at. The answer to the very interesting question puts forward by the referee deserves a special issue. In fact, looking at the literature, there is a huge discrepancy in the information available regarding, on the one hand, MC and microbiota which is very extensive and, on the other hand, that between TC and microbiota which is almost non-existent.